# A Parallel-Phase Demodulation-Based Distance-Measurement Method Using Dual-Frequency Modulation

**In-Gyu Jang** [ID]**, Sung-Hyun Lee and Yong-Hwa Park ***

Department of Mechanical Engineering, Korea Advanced Institute of Science and Technology (KAIST), Daejeon 34141, Korea; igjang@kaist.ac.kr (I.-G.J.); shleetop@kaist.ac.kr (S.-H.L.)
* Correspondence: yhpark@kaist.ac.kr; Tel.: +82-42-350-3235

**Abstract:** Time-of-flight (ToF) measurement technology based on the amplitude-modulated continuous-wave (AMCW) model has emerged as a state-of-the-art distance-measurement method for various engineering applications. However, many of the ToF cameras employing the AMCW process phase demodulation sequentially, which requires time latency for a single distance measurement. This can result in significant distance errors, especially in non-static environments (e.g., robots and vehicles) such as those containing objects moving relatively to the sensors. To reduce the measurement time required for a distance measurement, this paper proposes a novel, parallel-phase demodulation method. The proposed method processes phase demodulation of signal in parallel rather than sequentially. Based on the parallel phase demodulation, $2\pi$ ambiguity problem is also solved in this work by adopting dual frequency modulation to increase the maximum range while maintaining the accuracy. The performance of proposed method was verified through distance measurements under various conditions. The improved distance measurement accuracy was demonstrated throughout an extended measurement range (1–10 m).

**Keywords:** time-of-flight; AMCW; parallel phase demodulation; $2\pi$ ambiguity

## 1. Introduction

3D depth maps contain position and orientation information of objects unavailable in color images. Such 3D spatial information can be utilized in various types of engineering applications, such as unmanned vehicles, robots, human-recognition systems, medical applications, and even personal IT devices [1–7]. Specifically, in the field of machine vision, 3D depth maps are widely used for many purposes, including human motion capturing [5,6] and object recognition [7].

There are several ways to obtain a 3D depth map, such as stereoscopy [8,9], the structured light method [10,11], and the Time-of-flight (ToF) measurement method [12–16]. However, the stereoscopy and structured light methods require many cameras and incur excessive computational loads [13]. In addition, these methods can achieve high depth resolutions only when a relatively large triangulation base is obtained, meaning that the systems are often bulky [13]. Consequently, the ToF measurement method has been widely adopted as an alternative for 3D depth sensing due to its compactness and low calculation loads [12–16]. During ToF measurements [12–17], a pulsed or continuous optical signal is emitted to the object and the traveling time of the reflected light signal is measured. The traveling time, i.e., the ToF, is then converted into the distance.

The direct ToF measurement method [13,15,17], which uses a pulsed optical signal for emission, is one way to measure the ToF. After the emitted light is reflected from the object, the time-to-digital converter (TDC) returns the ToF of the reflected light. Although the direct ToF measurement method

is logically simple, a high-precision and expensive TDC capable of nanosecond-timed resolutions is required. Moreover, a relatively high-powered laser source is required for pulse modulation. To cope with this drawback, the use of the AMCW method is growing.

In the AMCW method, continuously periodic (a single sine waveform) light is generated by modulating an illumination source, such as laser diode [12–14]. After this modulated light is emitted onto the object, the reflected light is received by the detector, such as an avalanche photodiode [13]. In the AMCW method, the time delay between the emitted and reflected light signals is determined by corresponding phase delay, which is calculated from the cross correlation [12] between the reflected light signal and external demodulation signal. This indirect ToF method can achieve adequate accuracy and provides the compactness of the sensor system [12]. Many types of AMCW-based cameras have been developed and analyzed [13,16]. Among them, Mesa Imaging developed the SR4000/4500 cameras [13] which provide 3D depth maps at a resolution of 176 × 144. In addition to the SR series, many other companies and research teams have developed promising sensor devices for obtaining 3D depth information [13,16].

Performance improvements of ToF cameras have been pursued by enhancing the accuracy of distance measurement. Payne et al. [18] reduced the degree of nonlinearity in phase measurements using a phase-encoding approach, which improved the accuracy of distance measurements. Lee [19] solved the problem of motion blur stemming from rapid motions of objects using an altered form of the conventional four-bucket method. The use of multiple frequency components in the modulation and demodulation process to improve the performance has been studied. Gupta et al. [20] and Gutierrez-Barragan [21] tried the various combinations of multiple frequency components. In these works, optimal waveforms of modulation and demodulation signals were determined to improve distance accuracy. Payne et al. [22] and Bamgi et al. [23] adopted a dual-frequency modulation scheme to increase maximum range limited by the $2\pi$ ambiguity problem. The above mentioned AMCW ToF method utilizes sequential phase demodulation implemented in hardware. Consequently, the previous methods require sequential integration of phase samples, which results in processing time being required for the integration of $N$ phase samples per one-point distance measurement [12]. In most cases, four phase samples are used: 4-bucket algorithm [12–16].

Reducing the total processing time of the phase demodulation per pixel can improve the performance of distance measurement effectively with regard to motion blur suppression or accuracy improvement, especially in case of raster scanning type ToF sensors. With that knowledge, this paper contributes to decreasing the total processing time required to obtain a distance value using a novel parallel phase demodulation method. Compared to conventional ToF methods which calculate phase samples sequentially, in this paper, all phase samples are demodulated in parallel for one-point distance measurements. Although this method requires fast digitization of received light signal, this method needs only a single integration time for acquisition of samples. In addition to decreasing the processing time, the $2\pi$ ambiguity problem, which is the main cause of limitation of maximum measurable range, is solved by adopting dual-frequency modulation. By combining parallel phase demodulation and dual-frequency modulation, not only is the maximum range increased, but the accuracy at all distances is improved as well.

The remainder of this paper proceeds as follows: Section 2 explains the principles of the AMCW method. Section 3 mentions the $2\pi$ ambiguity problem and previous solutions [22,24,25]. Section 4 deals with details of the proposed method. Sections 5 and 6 show the experimental setup, results, and discussion. In Section 7, conclusions are presented.

## 2. The Principles of AMCW-Based ToF Measurement

General AMCW-based ToF measurement scheme is illustrated in Figure 1. The illumination source is modulated within the frequency range of 20–30 MHz usually. This modulated light is emitted onto the object, which reflects the modulated light back to the detector. Detected reflected light signal is demodulated by a demodulation pixel or by an optical shutter [12–14].

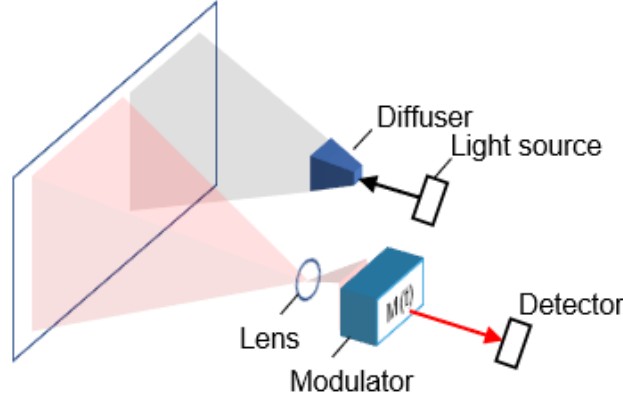

**Figure 1.** Scheme used by AMCW time of flight (ToF) cameras.

The harmonic emitted light signal $E(t)$, reflected light signal $R(t)$, and the $n$-th demodulation signal $M_n(t)$ are expressed below

$$E(t) = E_A \sin(2\pi ft) + E_{DC} \tag{1}$$

$$R(t) = R_A \sin(2\pi ft - \varphi_{ToF}) + R_{DC} \tag{2}$$

$$M_n(t) = M_A \sin(2\pi ft + \delta_n) + M_{DC} \tag{3}$$

$$\delta_n = \frac{2\pi(n-1)}{N}, \; n = 1, 2, 3, \ldots, N, \tag{4}$$

where $E_A$, $R_A$, and $M_A$ represent the amplitudes of the emitted light signal, the reflected light signal, and the demodulation signal, respectively. $f$ denotes the modulation frequency of all signals. $E_{DC}$, $R_{DC}$, and $M_{DC}$ represent the DC components of the emitted light signal, the reflected light signal, and the demodulation signal, respectively. $\varphi_{ToF}$ is the phase delay of the reflected light signal. In the demodulation signal, $\delta_n$ represents the $n$th phase shift for $n = 1, 2, \ldots, N$. The number of phase shifts, identical to the number of samples, is $N$.

During the demodulation process, the sample $I_m(\delta_n)$, which is the cross-correlation function between the reflected light signal and demodulation signal for one integration time, is calculated for $n$ = 1, 2, ... , $N$ in sequence [26]. The integration in the demodulation process can be expressed as

$$
\begin{aligned}
I_m(\delta_n) &= \int_0^{T_{int}} R(t)M_n(t)dt \\
&= \int_0^{T_{int}} \{R_A \sin(2\pi ft - \varphi_{ToF}) + R_{DC}\} \cdot \{M_A \sin(2\pi ft + \delta_n) + M_{DC}\}dt \\
&= \frac{R_A M_A T_{int}}{2} \cos(\varphi_{ToF} + \delta_n) + R_{DC}M_{DC}T_{int}, \; n = 1, 2, 3, \ldots N
\end{aligned}
\tag{5}
$$

In Equation (5), $T_{int}$ is the integration time required to calculate one sample. During the sample integration, sinusoidal terms which contain $f$ are generally negligible under the assumption that the inverse of $T_{int}$ is much smaller than $f$. With multiple samples, the phase delay between the emitted light signal and reflected light signal is calculated through a simple trigonometric calculation [26,27]. The distance is also determined using this phased delay accordingly. The results are given below

$$\varphi_{ToF} = \tan^{-1}\left( -\frac{\sum\limits_{n=1}^{N} I_m(\delta_n) \sin\left(\frac{2\pi(n-1)}{N}\right)}{\sum\limits_{n=1}^{N} I_m(\delta_n) \cos\left(\frac{2\pi(n-1)}{N}\right)} \right) \tag{6}$$

$$d = \frac{c}{4\pi f}\varphi_{ToF}, \tag{7}$$

where $c$ is the velocity of light and $d$ is the distance from the sensor to the object. More details can be found in the literature [18,22,24,27,28]. To measure a single distance, $N$ integration times are required for $N$ integration processes in Equation (5). By processing the phase demodulation in parallel, the total time for one-point distance measurement can be reduced. Details regarding the reduction of the total time for one-point distance measurement are presented in Section 4.

## 3. $2\pi$ Ambiguity Problem

The AMCW method has a limitation on the maximum range of the distance due to the $2\pi$ ambiguity problem. In the AMCW method, the maximum measurable distance, known as the unambiguous range, is expressed as follows

$$d_{\max} = \frac{c}{2f}. \tag{8}$$

The actual distance of an object at a distance exceeding the unambiguous range is expressed as

$$d_{act} = d + \frac{c}{2f} \times k, \ k = 0, 1, 2, 3, \ldots, \tag{9}$$

where $k$ is a non-negative integer. The $2\pi$ ambiguity relies on $k$, which cannot be determined for as long as single modulation frequency $f$ is used. If $k$ can be estimated, the actual distance can be found, and the $2\pi$ ambiguity problem can, thus, be solved. There are several ways to solve this problem, as described in the subsequent subsections.

### 3.1. Modulation Frequency Reduction

One way to mitigate the $2\pi$ ambiguity problem is to increase the unambiguous range by reducing the modulation frequency. It is obvious that the unambiguous range increases as the frequency decreases according to Equation (8). However, the uncertainty of the measured distance also increases as the modulation frequency decreases, as can be described using the standard deviation of distance measurement [12] as

$$\sigma_d = \frac{c}{4\sqrt{2}\pi f} \frac{\sqrt{B + N_{pseudo}}}{c_d A}, \tag{10}$$

where $c_d$ is referred to as demodulation contrast; $A$ is the amplitude of the sample $I_m(\delta_n)$ in Equation (5) ($\frac{R_A M_A T_{int}}{2}$); $B$ is DC component of sample $I_m(\delta_n)$ in Equation (5) ($R_{DC}M_{DC}T_{int}$), which partially stems from the DC component of the demodulation signal $M_{DC}$; and $N_{pseudo}$ represents other noise sources which are not related to the DC component [12]. According to this equation, a reduction of the modulation frequency increases the standard deviation. Therefore, the unambiguous range and the accuracy of the distance measurement have a trade-off relationship.

### 3.2. Sequential Dual-Frequency Modulation Approach

Another method that can be used to solve the $2\pi$ ambiguity problem is dual-frequency modulation. Dorrington et al. [24] used two different modulation frequencies to solve the measurement ambiguities. In this method, two measured distances are obtained for each modulation frequency sequentially using the AMCW method. The actual distance is then calculated using these two measured distances [24] as follows. Figure 2 shows the overall scheme when using these two modulation frequencies sequentially.

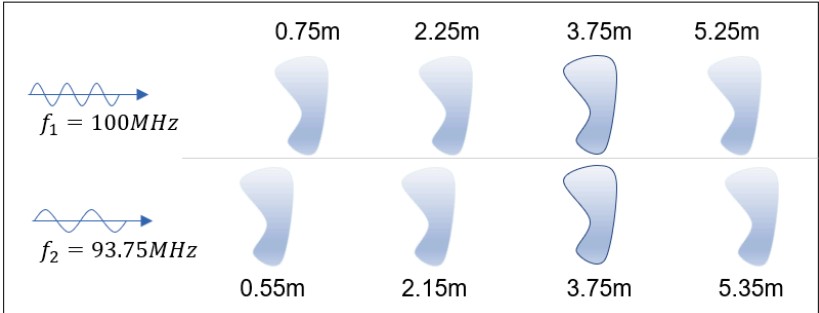

**Figure 2.** Sequential dual-frequency modulation scheme.

When using this method, the unambiguous range, which is increased, can be given by

$$d_{\max} = \frac{c}{2f_E} = \frac{c}{2 \times \gcd(f_1, f_2)}, \tag{11}$$

where $f_E$ is the effective frequency, which is identical to the greatest common division of the two frequencies. In addition, $f_1$ and $f_2$ represent the two modulation frequencies.

The advantage of this method is that the calculations are simple and it uses a high frequency to measure the distance, which improves the accuracy of the measurement. However, the total time for one-point distance measurement is doubled due to the sequential dual-frequency modulation process [22].

### 3.3. Simultaneous Dual Frequency Modulation with Sequential Demodulation

In order to overcome the disadvantages of the sequential dual-frequency modulation method, there is an alternative method which superposes each modulation frequency into a single modulation signal but demodulates it with different frequencies ($f_1$ and $f_2$) sequentially to get a single phase image [22]. By repeating the procedure using the different initial phase shift, $\delta_n$, $n = 1, 2, \ldots, N$, $N$ different samples are obtained to extract actual distance. Figure 3 is schematic of simultaneous dual frequency modulation with sequential demodulation, which shows waveform of modulation and demodulation signals [22].

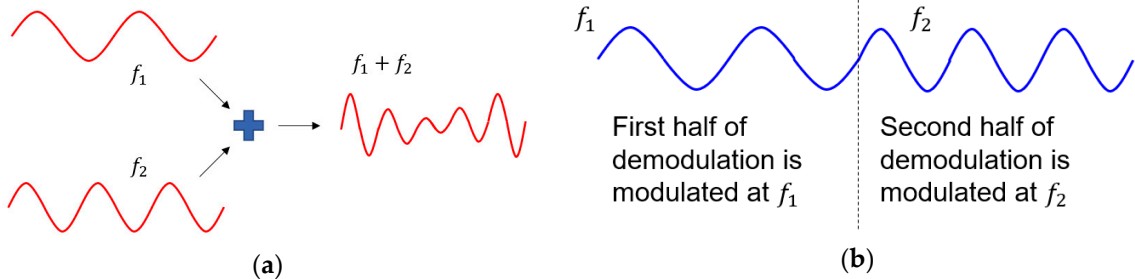

**Figure 3.** Schematic of simultaneous dual frequency modulation with sequential demodulation. (**a**) Superposition of two modulation frequencies to modulate the illumination source; (**b**) demodulation signal separated into two parts in sequence.

The main advantage of this method is that it does not significantly increase the total time for one actual distance measurement compared to the single-frequency modulation method described in Section 3.1 [22]. One disadvantage of this method, however, is that the demodulation signal must change over time. Moreover, it requires more than five-samples to obtain a single distance [22].

## 4. Proposed Parallel Phase Demodulation Using Dual Frequencies

The dual-frequency modulation scheme based on parallel phase demodulation is described in this section. The main advantage of the proposed method is that the demodulation processing time is nearly identical to one integration time regardless of the number of samples per distance measurement. This is possible because the proposed method uses parallel phase demodulation, meaning that all samples are calculated in parallel.

### 4.1. Parallel Phase Demodulation Method

The conventional AMCW method demodulates the optical signal reflected from an object using hardware such as an optical shutter, demodulation pixel, or an image intensifier [12,14,24]. Specifically, only one demodulation signal can be used per integration time; therefore, demodulation must be done sequentially. The parallel phase demodulation method proposed in this paper undertakes computational demodulation using software rather than hardware. The overall schematics of the sequential and parallel phase demodulations are shown in Figure 4.

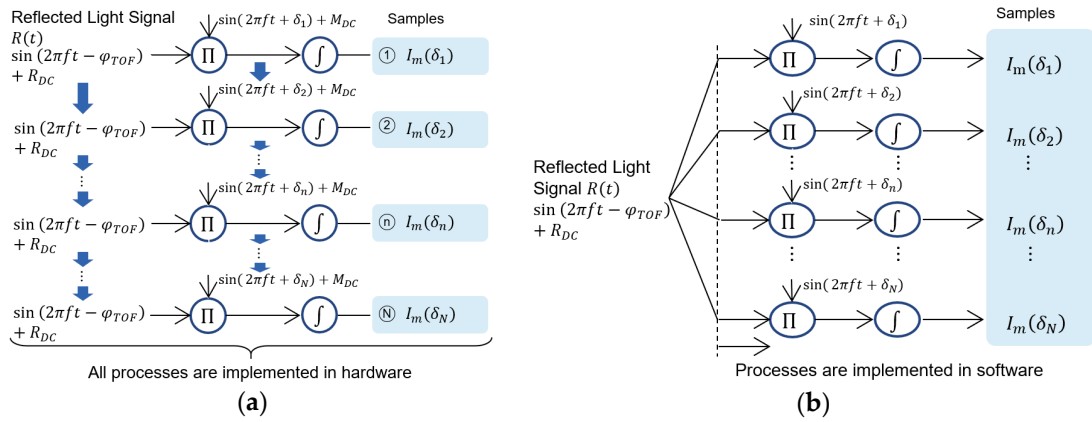

**Figure 4.** Comparison of sequential phase demodulation and parallel phase demodulation methods. (**a**) Sequential demodulation process of the conventional AMCW method; (**b**) parallel demodulation process of the proposed method.

Figure 4a shows the sequential demodulation process of conventional AMCW method. In the conventional method, the samples $I_m(\delta_n)$, $n = 1, 2, \ldots, N$, are sequentially obtained for distance calculations. The sequential processing is inevitable in this case because it adopts a demodulation process in its hardware [12,14,24]. Figure 4b shows the proposed parallel phase demodulation process. Reflected light signal $R(t)$ is measured and converted to digital format; then, the samples $I_m(\delta_n)$, $n=1, 2, \ldots, N$, are calculated using $N$ number of demodulation signals in parallel compotation. The proposed method requires $1/N$ total processing time per single distance measurement compared to that of sequential demodulation. In addition, the proposed method provides ideally zero DC components of demodulation $M_{DC}$, which improves distance accuracy compared to that of the conventional method, as described in Equation (10). The proposed method demodulates digitized reflected light; and in software, positive and negative values of the demodulation signal, for example, ideal sine wave functions with zero DC components, can be used. Whereas the conventional method based on hardware demodulation uses a physical demodulation signal [12,14,24], which has a positive value for all time, a sine wave form with positive, non-zero DC component value can be used.

Another advantage of parallel phase demodulation is that it can reduce unwanted noise effects during the calculation process. To verify this, the overall procedure of the parallel phase demodulation

method is described mathematically. First, the reflected light signal $R(t)$ and the demodulation signal $M_n(t)$ can be expressed as follows

$$R(t) = R_A \sin(2\pi f t - \varphi_{ToF}) + R_{DC} \tag{12}$$

$$M_n(t) = M_A \sin(2\pi f t + \delta_n), \tag{13}$$

where all related parameters are identical to those in Section 2, and as mentioned above, the demodulation signal in Equation (13) has zero DC components. The $n$th sample $I_m(\delta_n)$, which is calculated by demodulating the reflected light signal, can then be expressed as

$$I_m(\delta_n) = \int_0^{T_{int}} R(t) M_n(t) dt. \tag{14}$$

This equation can be rewritten as shown below

$$
\begin{aligned}
I_m(\delta_n) &= \int_0^{T_{int}} \{R_A \sin(2\pi f t - \varphi_{ToF}) + R_{DC}\}\{M_A \sin(2\pi f t + \delta_n)\} dt \\
&= \int_0^{T_{int}} \left\{ \frac{R_A M_A}{2} \cos(\varphi_{ToF} + \delta_n) - \frac{R_A M_A}{2} \cos(4\pi f t - \varphi_{ToF} + \delta_n) + R_{DC} M_A \sin(2\pi f t + \delta_n) \right\} dt \\
&= \frac{R_A M_A T_{int}}{2} \cos(\varphi_{ToF} + \delta_n)
\end{aligned}
\tag{15}
$$

Hence, there is no DC component on $M_n(t)$, which is related to photon shot noise [12]. With the parallel phase demodulation method, it is shown that such DC terms on $M_n(t)$ can be excluded because all demodulation signals are implemented in the software program, as mentioned above. Moreover, in the sample $I_m(\delta_n)$, there is no DC term related to $R_{DC}$, which includes the external light. This indicates that the proposed method can reduce the effects of external light on the accuracy of the measurement. In summary, the proposed parallel phase demodulation method can reduce both the total time for one distance measurement and the noise effects.

*4.2. Dual-Frequency Modulation Based on Parallel Phase Demodulation*

In order to solve the $2\pi$ ambiguity problem, the conventional dual-frequency modulation [24,25] method is combined with parallel phase demodulation. By combining these two methods, the demodulation process is especially reformed, which is different with sequential demodulation process mentioned in Section 3.3.

The mathematical description is as follows. First, the emitted light signal $E(t)$ is modulated simultaneously using two frequencies. In this case, the emitted light signal $E(t)$ and the reflected light signal $R(t)$ can be expressed as follows:

$$E(t) = E_1 \sin(2\pi f_1 t) + E_2 \sin(2\pi f_2 t) + E_{DC} \tag{16}$$

$$R(t) = R_1 \sin(2\pi f_1 t - \varphi_1) + R_2 \sin(2\pi f_2 t - \varphi_2) + R_{DC}, \tag{17}$$

where $E_1$, $E_2$, $R_1$, and $R_2$ represent the amplitude of the $f_1$ component of the emitted light signal, the amplitude of the $f_2$ component of the emitted light signal, the amplitude of the $f_1$ component of the reflected light signal, and the amplitude of the $f_2$ component of the reflected light signal, respectively. Above $\varphi_1$ and $\varphi_2$ are, correspondingly, the phase shifts of the reflected light signal for each frequency component. Likewise, the demodulation signal is also generated using two frequencies simultaneously. The demodulation signal can be expressed as

$$M_n(t) = M_1 \sin(2\pi f_1 t + m\delta_n) + M_2 \sin(2\pi f_2 t + l\delta_n) \tag{18}$$

$$\delta_n = \frac{2\pi(n-1)}{N}, \ n = 1, 2, 3, \ldots, N \tag{19}$$

$$m = \frac{f_1}{f_e}, \ l = \frac{f_2}{f_e}, \ f_e = \gcd(f_1, \ f_2), \tag{20}$$

where $M_1$ and $M_2$ represent the amplitude of the $f_1$ component of the demodulation signal and the amplitude of the $f_2$ component of the demodulation signal, respectively. $f_e$ is the effective frequency, which is identical to the greatest common division of $f_1$ and $f_2$. The demodulation signal does not have a DC component because demodulation is done using software, as opposed to other methods in the literature [22,24,25]. Moreover, as expressed by Equation (18), the demodulation signal is also generated as the sum of the dual-frequency signal, which is also one of the major differences compared to earlier approaches [22,24,25]. Using the signals above, the $2\pi$ ambiguity problem is revisited by referring to previous methods [22,24,25].

When the phase shift of the demodulation signal is 0, sample $I_m(0)$ can be expressed as follows

$$
\begin{aligned}
I_m(0) &= \int_0^{T_{\text{int}}} R(t) M_0(t) dt \\
&= \int_0^{T_{\text{int}}} \{R_1 \sin(2\pi f_1 t - \varphi_1) + R_2 \sin(2\pi f_2 t - \varphi_2) + R_{DC}\}\{M_1 \sin(2\pi f_1 t) + M_2 \sin(2\pi f_2 t)\} dt \\
&= \int_0^{T_{\text{int}}} \left\{\frac{R_1 M_1}{2} \cos(\varphi_1) - \frac{R_1 M_1}{2} \cos(4\pi f_1 t - \varphi_1) + \frac{R_2 M_2}{2} \cos(\varphi_2) - \frac{R_2 M_2}{2} \cos(4\pi f_2 t - \varphi_2)\right\} dt \\
&+ \int_0^{T_{\text{int}}} \left\{\frac{R_2 M_1}{2} \cos(2\pi(f_2 - f_1)t + \varphi_1) - \frac{R_2 M_1}{2} \cos(2\pi(f_2 + f_1)t - \varphi_1)\right\} dt \\
&+ \int_0^{T_{\text{int}}} \left\{\frac{R_1 M_2}{2} \cos(2\pi(f_1 - f_2)t + \varphi_2) - \frac{R_1 M_2}{2} \cos(2\pi(f_1 + f_2)t - \varphi_2)\right\} dt
\end{aligned}
\tag{21}
$$

Because the inverse of $T_{\text{int}}$ is much smaller than $f_1$, $f_2$, $|f_1 - f_2|$ and $f_1 + f_2$, the terms which contain the $f_1$, $f_2$, $|f_1 - f_2|$ and $f_1 + f_2$ can be neglected. Thus, sample $I_m(0)$ can be expressed as

$$I_m(0) = \frac{R_1 M_1 T_{\text{int}}}{2} \cos(\varphi_1) + \frac{R_2 M_2 T_{\text{int}}}{2} \cos(\varphi_2). \tag{22}$$

Likewise, if the phase shift $\delta_n$ is applied to the demodulation signal, sample $I_m(\delta_n)$ can be expressed as

$$I_m(\delta_n) = \frac{R_1 M_1 T_{\text{int}}}{2} \cos(\varphi_1 + m\delta_n) + \frac{R_2 M_2 T_{\text{int}}}{2} \cos(\varphi_2 + l\delta_n). \tag{23}$$

Using angle sum identities and the orthogonality of trigonometric functions, the following equations are satisfied.

$$
\begin{aligned}
\sum_{n=1}^{N} I_m(\delta_n) \cos(m\delta_n) &= \frac{R_1 M_1 T_{\text{int}} N}{2} \cos(\varphi_1) \\
\sum_{n=1}^{N} I_m(\delta_n) \sin(m\delta_n) &= -\frac{R_1 M_1 T_{\text{int}} N}{2} \sin(\varphi_1) \\
\sum_{n=1}^{N} I_m(\delta_n) \cos(l\delta_n) &= \frac{R_2 M_2 T_{\text{int}} N}{2} \cos(\varphi_2) \\
\sum_{n=1}^{N} I_m(\delta_n) \sin(l\delta_n) &= -\frac{R_2 M_2 T_{\text{int}} N}{2} \cos(\varphi_2)
\end{aligned}
\tag{24}
$$

The distance using each frequency can then be calculated as shown below

$$d_1 = \frac{c}{4\pi f_1}\varphi_1 = \frac{c}{4\pi f_1}\tan^{-1}\left(-\frac{\sum\limits_{n=1}^{N} I_m(\delta_n)\sin\left(m\frac{2\pi(n-1)}{N}\right)}{\sum\limits_{n=1}^{N} I_m(\delta_n)\cos\left(m\frac{2\pi(n-1)}{N}\right)}\right)$$

$$d_2 = \frac{c}{4\pi f_2}\varphi_2 = \frac{c}{4\pi f_2}\tan^{-1}\left(-\frac{\sum\limits_{n=1}^{N} I_m(\delta_n)\sin\left(l\frac{2\pi(n-1)}{N}\right)}{\sum\limits_{n=1}^{N} I_m(\delta_n)\cos\left(l\frac{2\pi(n-1)}{N}\right)}\right)$$
(25)

Using $d_1$, $d_2$, and Equation (9), $k_1$ and $k_2$, which are the integers expressed in Equation (9) for each frequency, are determined as follows

$$(k_1, k_2) = \underset{k_1, k_2}{\operatorname{argmin}}\left\{\left(d_1 + \frac{c}{2f_1}k_1\right) - \left(d_2 + \frac{c}{2f_2}k_2\right)\right\}.$$
(26)

The values of $k_1$ and $k_2$ can be determined using the modified Chinese remainder theorem [29]. An error $e$ can arise when calculating $k_1$ and ... The standard deviation of the error, i.e., $\sigma_e$, and the probability of the occurrence of an error, i.e., $P_e$, can be expressed as follows [25]

$$\sigma_e \propto \sqrt{\left(\frac{f_1}{f_e}\right)^2 \times \frac{1}{(R_2 M_2)^2} + \left(\frac{f_2}{f_e}\right)^2 \times \frac{1}{(R_1 M_1)^2}}$$
(27)

$$P_e = 1 - erf\left(\frac{1}{2\sqrt{2}\sigma_e}\right).$$
(28)

The above shows that the higher the frequencies which are combined, the higher the probability of the occurrence of an error becomes [25]. After calculating $k_1$ and $k_2$, the actual distance is calculated by interpolating the two measured distances $d_1 + \frac{c}{2f_1}k_1$ and $d_2 + \frac{c}{2f_2}k_2$ [25].

$$d_{act} = w\left(d_1 + \frac{c}{2f_1}k_1\right) + (1-w)\left(d_2 + \frac{c}{2f_2}k_2\right), \ 0 \le w \le 1.$$
(29)

The optimum value of $w$ which minimizes the standard deviation of the actual distance is expressed as shown below [25].

$$w = \frac{1}{1 + \left(\frac{f_2}{f_1}\right)^2\left(\frac{R_1 M_1}{R_2 M_2}\right)^2}.$$
(30)

The main advantage of the method proposed above is that the measurement time does not change, even if numerous samples are used for the distance measurement. In Sections 3.2 and 3.3, the sequential dual-frequency method's implementation is described; it is shown that an increased number of samples can reduce the measurement accuracy if total measurement time remains. However, the proposed method does not have such a drawback. Moreover, as mentioned in Section 4.1, there is no DC component for sample $I_m(\delta_n)$, as indicated in Equation (23). This makes it possible to remove the effects of external light on the accuracy of the measurement.

## 5. Experimental Setup

Figure 5a shows the whole distance measurement system. The distance measurement system consists of five components: a monitor for displaying the results of the experiment, an analyzer for computing all calculations in real time, a laser controller, a laser diode, and an avalanche photodiode. Figure 5b shows the optical components.

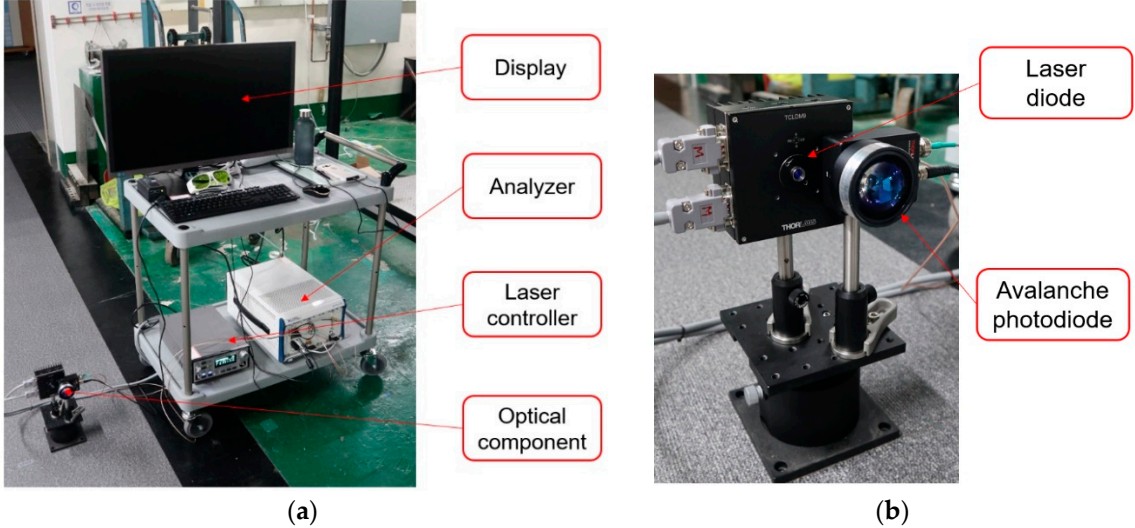

**Figure 5.** (**a**) The whole distance measurement system; (**b**) optical components of the system.

The M9-852-0100 laser diode by Thorlabs Inc., which generates infrared light at a wavelength of 852 nm, is used as the illumination source. The APD 120 by Thorlabs Inc. is used as the detector, which converts the reflected light signal into a voltage signal. The PXI system by National Instruments Inc. is used to control the measurement system, collect the measurement signal, and process the signals. The PXI system consists of a PXIe-5160 digitizer module, a PXIe-5423 arbitrary function generator module, and a PXI-8880 controller module. The PXIe-5160 fast digitizer module collects the measured voltage signal from the avalanche photodiode with a sampling frequency of 625 MHz. The PXIe-5423 module generates a waveform for modulating the laser intensity and for generating the demodulation signal. The PXI-8880 controller module controls the entire measurement system and processes the samples. All modules are combined in a PXI-1082 chassis, which provides the power, cooling, and communication buses for the PXI modules.

Figure 6 shows the overall signal flows of the measurement system. The laser controller generates a DC component for the signal, and the function generator generates a modulation signal. Part of the modulation signal is connected to one channel of the digitizer to be used as the demodulation signal. The remaining generated modulation signal is combined with the DC component of the laser controller and used to modulate the laser diode, which emits the optical signal onto the object. The emitted light signal is then reflected and detected by the avalanche photodiode, after which the avalanche photodiode transmits the measured voltage signal to the digitizer through another channel. The digitizer collects input signals and transmits them to the controller. The controller calculates the distance using the signals received. One distinct advantage of this system setup is that the modulation signal and demodulation signal can be generated using only one function generator.

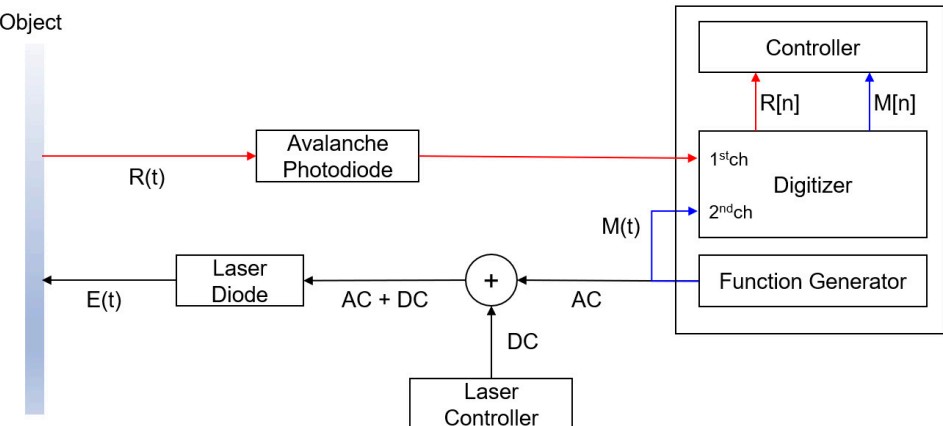

**Figure 6.** Signal flows of the measurement system.

## 6. Results

### 6.1. Results from Single-Frequency Modulation

In this section, using the parallel phase demodulation method, the standard deviations with respect to the object distance are analyzed. The conditions of the experiment are determined by two independent variables: the modulation frequency and the modulation voltage amplitude. The measured distance ranged from 1 m to 4.5 m at intervals of 0.5 m. For each distance, 10,000 repeated distance measurements were conducted to evaluate the standard deviations of the result. The remaining factors were fixed; i.e., the sampling frequency at 625 MS/s, the number of samples for a single distance measurement at 4, and the integration time at 9.6 µs.

Figure 7 shows the results of the experiment for different modulation frequencies. The modulation frequencies were 31.25, 15.625, and 7.8125 MHz. The modulation voltage amplitude was fixed at 12 V. Figure 7 shows that as the modulation frequency increased, the standard deviation decreased, which can be described by Equation (10). For the case of 7.8125 MHz modulation frequency, the maximum value of the standard deviation was about 0.05 m. On the other hand, for 31.25 MHz modulation frequency, the maximum value of the standard deviation was lower than 0.01 m.

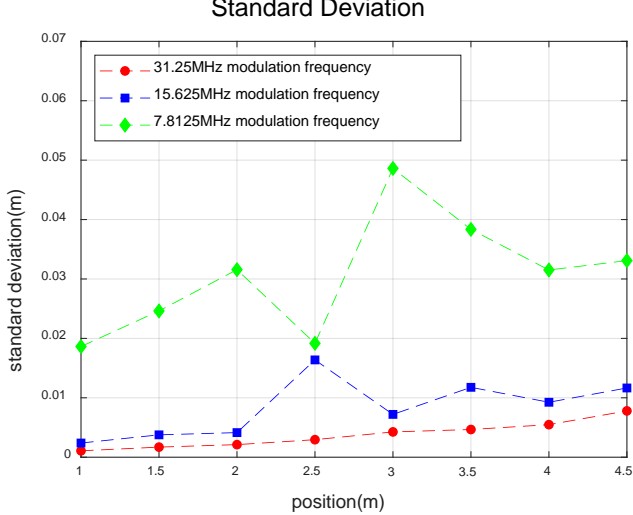

**Figure 7.** Standard deviation per modulation frequency.

Figure 8 presents the results of the experiments for different modulation voltage amplitudes, 12, 8, and 4 V. The modulation frequency was fixed at 31.25 MHz. It shows that as the modulation voltage amplitude increases, the standard deviation decreases. The maximum value of standard deviation is

lower than 0.02 m. A monotonically increasing trend of standard deviation is shown for all cases of modulation voltage amplitude in general, which can be described by Equation (10).

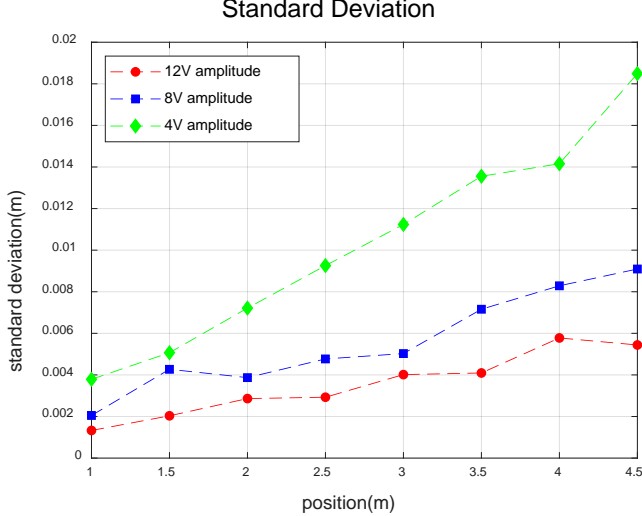

**Figure 8.** Standard deviation for each modulation voltage amplitude.

Performances of the proposed method and two commercial cameras utilizing conventional sequential AMCW [30], are compared in Table 1 in terms of distance standard deviation versus distance and integration time. The performance parameters of commercial cameras were taken from a reference paper [30]. The relatively very short integration time of the proposed method was chosen to be 9.6 μs, which is suitable for raster scanning type distance measurements. From Table 1, it is obvious that the proposed method can measure a single distance in a very short integration time. The proposed method presents relatively low standard deviation contributed by no DC components in the demodulation process, as shown in Equations (10) and (13).

**Table 1.** Standard deviations of conventional cameras and the proposed method.

| Comparison Item | 1 m-Distance Object | | 1.3 m-Distance Object | | 1.6 m-Distance Object | |
|---|---|---|---|---|---|---|
| | Integration Time | Standard Deviation | Integration Time | Standard Deviation | Integration Time | Standard Deviation |
| SR-4000 [30] | 3.5 ms | 0.016 m | 8.5 ms | 0.008 m | 16.75 ms | 0.008 m |
| CamCube3.0 [30] | 0.05 ms | 0.008 m | 0.1 ms | 0.010 m | 0.25 ms | 0.009 m |
| This proposed method | 9.6 μs | 0.0011 m | 9.6 μs | 0.0015 m | 9.6 μs | 0.0018 m |

*6.2. Results from Dual-Frequency Modulation*

In this section, using dual-frequency modulation based on parallel phase demodulation, the bias error and the standard deviation are investigated. The conditions of the experiment were determined by the combination of the two frequencies. In this case, combinations of 31.25 and 34.375 MHz, 21.875 and 34.375 MHz, and 3.125 and 34.375 MHz were used. The unambiguous ranges of each frequency combination are shown in Table 2 which can be calculated using Equations (8) and (11). The distance was measured from 1 to 10 m at intervals of 0.5 m. For each distance, 10,000 measurements were conducted. The remaining factors were fixed; i.e., the sampling frequency at 625 MS/s, the modulation voltage amplitude at 12 V, eight samples for one-point distance measurement, and an integration time of 9.6 μs.

**Table 2.** Unambiguous ranges of each frequency combination

| Modulation Frequency | Unambiguous Range |
|---|---|
| 34.375 MHz | 4.363 m |
| 31.25 MHz | 4.8 m |
| 21.875 MHz | 6.857 m |
| 3.125 MHz | 48 m |
| 31.25 MHz + 34.375 MHz | 48 m |
| 21.875 MHz + 34.375 MHz | 48 m |
| 3.125 MHz + 34.375 MHz | 48 m |

Figure 9a shows that the bias error when using only single low frequency is much larger than that when using dual frequencies. Specifically, the maximum value of bias error when using single low frequency is higher than 0.4 m. On the other hand, the maximum value of bias error for dual-frequency cases is about 0.1 m. Figure 9b shows that the standard deviation when using only single low frequency is much larger than that when using dual frequencies in general. It should be noted that as higher frequencies are combined, the standard deviation becomes smaller. This trend is also shown in earlier work [25]. However, in case of combination of 31.25 and 34.375 MHz, there is an abrupt increase in the standard deviation at around 9.0 m, which was predicted in Equations (27) and (28) [25]. Consequently, as shown in Figure 9b, the best combination of dual frequencies is the intermediate frequency and high frequency; i.e., 21.875 and 34.375 MHz.

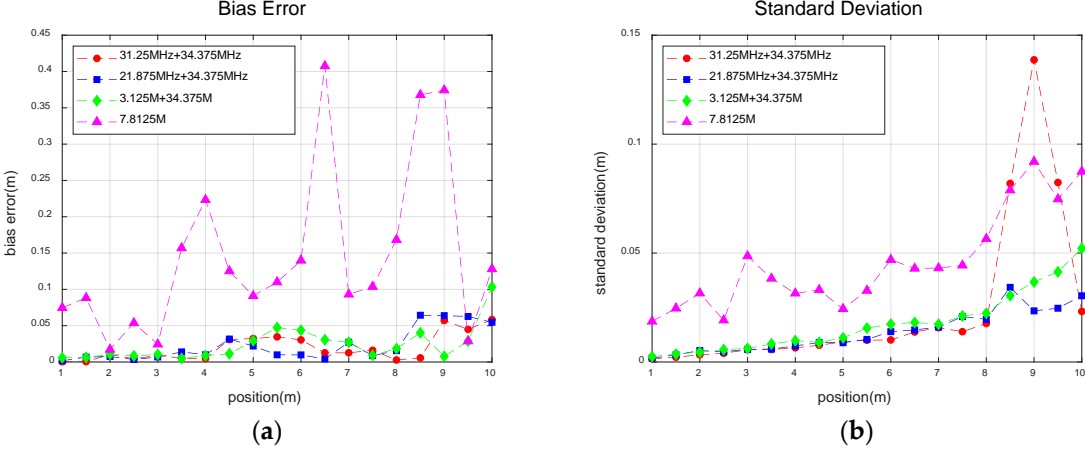

**Figure 9.** (**a**) Bias error for each frequency combination; (**b**) standard deviation for each frequency combination.

## 7. Conclusions

In this paper, an AMCW-based ToF measurement method using parallel phase demodulation and dual-frequency modulation is proposed. The performance of this proposed method was also verified through experimental results. According to the results, the proposed method presents relatively low standard deviation of distance measurement due to zero DC components of demodulation signal. In addition to improvement of accuracy, time for single distance measurement can be reduced by the proposed method. Moreover, using the dual-frequency modulation, the proposed method can extend unambiguous range effectively, as shown in experimental results. The proposed method will be applied to a compact scanning-type 3D sensor for robot applications in the author's group.

**Author Contributions:** Conceptualization, I.-G.J.; methodology, I.-G.J.; software, I.-G.J.; validation, I.-G.J. and S.-H.L.; formal analysis, I.-G.J. and S.-H.L.; investigation, I.-G.J. and S.-H.L.; resources, I.-G.J. and S.-H.L.; data curation, I.-G.J. and S.-H.L.; writing—original draft preparation, I.-G.J. and S.-H.L.; writing—review and editing, I.-G.J., S.-H.L., and Y.-H.P.; visualization, I.-G.J. and S.-H.L.; supervision, Y.-H.P.; project administration, Y.-H.P.; funding acquisition, Y.-H.P. All authors have read and agreed to the published version of the manuscript.

**Funding:** This research received no external funding.

**Acknowledgments:** This work was supported by the "Human Resources Program in Energy Technology" project of the Korea Institute of Energy Technology Evaluation and Planning (KETEP), granted financial resources from the Ministry of Trade, Industry, and Energy of the Republic of Korea (number. 20184030202000). This work was also supported by the National Research Foundation of Korea (NRF) grant funded by the Korea government (MSIT) (number 2017R1A2B2010759).

**Conflicts of Interest:** The authors declare no conflict of interest.

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
