# Peer review of "A Parallel-Phase Demodulation-Based Distance-Measurement Method Using Dual-Frequency Modulation"

_applsci, doi:10.3390/app10010293_

Round 1

Reviewer 1 Report

The paper describes a method for multi frequency depth measurements with a frequency domain capture scheme. It also proposes a method for parallel capture of the different modulation phases required for a frequency domain time-of-flight measurement. The paper is structured well and reasonably well written. It ignores some of the prior art and the proposed method has some severe short commings compared to other similar sensors. I cannot recommend publication of the paper in it's current form. In particular my concerns are:

1. The parallel capture approach requires knowledge of the reflected light function R(t). As is described in the paper the modulation and integration step (Equation 14) is performed by hardware in existing systems. This means that the hardware needs a detector with modulated gain or a frequency mixer to multiply R and M and then a slow detector and a slow digitization and readout circuit to read out I. The fact that we can rely on a analog gain modulation M and slow readout electronics is the main appeal of frequency domain measurements. It is the reason why they are preferred over direct time domain measurements that need to be able to sample the fast changing reflected light R(t) directly. The parallel capture method proposed here relies on fast digitization of R(t) and subsequent digital computation of the mixing (Equation 14). This means that most of the advantage of the Fourier domain method is gone and the proposed method will require detection hardware that is just as complex and expensive as the direct time domain method. There is still some potential advantage in the similar illumination. Using continuously modulated lasers rather than pulses may in some cases allow us to use more light.
I think the use of this method have to be better motivated and differences and drawbacks to existing methods should be mentioned and analyzed in the paper.

2. Simultaneous multi frequency measurements have been studied before: A theory for modeling modulation and demodulation frequencies with multiple frequencies has been published in

Mohit Gupta, Andreas Velten, Shree K. Nayar, and Eric Breitbach. 2018. What Are Optimal Coding Functions for Time-of-Flight Imaging?. ACM Trans. Graph. 37, 2, Article 13 (February 2018), 18 pages. DOI: https://doi.org/10.1145/3152155

Felipe Gutierrez-Barragan, Syed Azer Reza, Andreas Velten, Mohit Gupta; Practical Coding Function Design for Time-Of-Flight Imaging. The IEEE Conference on Computer Vision and Pattern Recognition (CVPR), 2019, pp. 1566-1574

The method proposes Hamiltonian modulation and demodulation functions that are effectively a superposition of more than two frequencies. The presented method is a special case within this theory that uses only two frequencies. The relation between the two contributions should be discussed.

In addition to these main points I have some smaller comments.
- The manuscript is readable, but the grammar and sentence structure is still incorrect in many places. I recommend having it checked again for correct grammar.
- The explanation of the method and measurement is confusing (Section 4). It should be made clearer what is captured in the different methods and how it is used. I.e. in this method we capture R and not I.

I believe the problems with the manuscript (items 1 and 2) have to be addressed before publication can be considered.

Reviewer 2 Report

DDear Authors,   This manuscript reports a method for high-accuracy distance measurement based on time-of-flight technology. In the proposed method, parallel phase modulation is employed   The authors mentioned that the proposed method can extend measured distance range in comparison with the conventional method for the sequence phase modulation.  The proposed method is experimentally estimated in the paper.    I think the paper should improve some descriptions.    1. In Lines 120, A is defined as the amplitude of the measured signal.     I cannot understand the difference "A" with  "M" defined in Line 86.     In other words, I do not understand the difference the measured signal and demodulation one. The authors should revise the paper to clarify the difference.   2.  In subsection 3.3, sequence demodulation is introduced. However, readers cannot understand the procedure in the demodulation. At least, the authors should explain the detail for the procedure to obtain the demodulation signal in Fig. 3(b) from the signal in Fig. 3 (a).   3. In the proposed method described in Fig. 4(b), measured signal is divided for parallel phase modulation. I think the range of each divided signal is decreased  in comparison with the measured one. If the divided signals may be amplified, SNR of the signals are worse than the measured one. The authors should explain the demerit of the proposed parallel modulation and discuss on the effects of the demerits. I judge that the problem is the most important for revision.   4.  In Line 272 the author write as follows:    "10,000 measurements are conducted".       Is "10,000" is the number of "t" or "n"?     This sentence should be re-written for much clearer explanation.        5. In last, the authors should discuss on experimental verification.     Especially, comparison with the conventional methods in terms of accuracy and range of distance quantitatively is considered to be required in the revised paper.     Regards,    

Round 2

Reviewer 1 Report

My main concerns have been addressed and the manuscript now puts the work in context appropriately. I do believe that the proposed method does away with most of the advantages of modulation based ToF methods and therefore the applications of the work are limited. However, since there are not scientific concerns, I am okay with publication.

The paper still needs significant editing for grammar and language.

Reviewer 2 Report

Dear Authors,

This manuscript a revised paper to report a method for high-accuracy distance measurement based on time-of-flight technology.
In the proposed method, parallel phase modulation is employed
The authors mentioned that the proposed method can extend measured distance range in comparison with the conventional method for the sequence phase modulation.
The proposed method is experimentally estimated in the paper.

In the previous review, 5 problems are mentioned to brush up the paper.
For 1, 4, and 5, the authors rewrite the sentences and give helpful explanations to understand the effectiveness of the proposed method.

For No. 2 and 3, on the other hand, I judge that the authors should revise the paper.

-----------------
For No.2 (This is simple problem), The authors do not refer Fig. 3 in the manuscript.
They should refer the important figure.
----------------

---------------------
For No.3, the authors explain that "the proposed parallel method can make the DC component of Mn(t) to zero". I cannot understand why the only proposed method can make delete the DC term whereas the conventional one cannot that. Why software program implementation is useful for the deletion?
Explanation for that is not enough to understand.
The authors should clearly explain in detail.

Note that, I can understand that deletion of DC terms is much effective for high accuracy distance measurement.
--------------------------------

Yours sincerely,
